# Investigating room temperature ferroelectric nematogens and their structure-property relationships

Naila Tufaha [1], Gytis Stepanafas[1], Ewan Cruickshank [1,3] ✉,
Damian Pociecha [2], Ewa Gorecka [2], John M. D. Storey [1],
Rebecca Walker [1] & Corrie T. Imrie [1,4]

The recent discovery of proper ferroelectricity in low molar mass liquid crystals has made the quest for identifying materials which exhibit the phase transition at room temperatures of upmost importance. Here, the synthesis and characterization of twelve different series of ferroelectric nematogens are reported based around the RM734 template structure. These compounds are varied by changing the terminal group; the position of the lateral alkyloxy chain along the mesogenic core; the number of lateral alkyloxy groups present; the degree of fluorination; and the length of the lateral alkyloxy chain. Such systematic modifications allow for a detailed analysis of the structure-property relationships for the ferroelectric nematic phase. All 70 of the compounds reported exhibit the ferroelectric nematic phase, with most exhibiting direct $N_F$-I transitions. Despite these transitions in general being monotropic, the vast majority of these compounds can be supercooled to room temperature with many showing a high degree of stability. Most notably, however, 19 of the reported compounds exhibit a transition to the $N_F$ phase below 30 °C. Only a single pure compound to date has exhibited a transition to the $N_F$ phase in this temperature range, UUQU-4N, so this is a dramatic increase in the library of compounds which exhibit the $N_F$ transition around room temperature. By understanding the molecular structure features leading to lowering these I-$N_F$ transition temperatures we move closer to an enantiotropic room temperature ferroelectric nematogen suitable for devices. Together the phase stability and the viscosity of ferroelectric materials, which governs polarization switching dynamics, are tuned by molecular design, allowing control over the balance between switching speed and long-term retention of the polarized state.

The simplest and the most widely used liquid crystal phase is the nematic, N, phase, which has been exploited in liquid crystal display technology for over 50 years[1]. More recently, a variant of the nematic phase, the ferroelectric nematic, $N_F$, phase has attracted significant research interest due to its spontaneous polarity coupled with fluidity. The high sensitivity of the $N_F$ phase to electrical and magnetic fields makes it a real candidate for applications such as new electro-optical devices, high-power supercapacitors and many other technologies[2–4]. The existence of this phase was predicted over 100 years ago by Born[5], who suggested that if molecules had a sufficiently

[1]Department of Chemistry, University of Aberdeen, Old Aberdeen, UK. [2]Faculty of Chemistry, University of Warsaw, Warsaw, Poland. [3]Present address: School of Pharmacy, Applied Sciences and Public Health, Robert Gordon University, Aberdeen, UK. [4]Deceased: Corrie T. Imrie. ✉e-mail: e.cruickshank2@rgu.ac.uk

large molecular dipole moment, the interactions between them could drive the formation of the nematic phase and, provided those interactions were strong enough to withstand thermal fluctuations, this phase would be ferroelectric in nature. While many materials with large longitudinal dipoles were synthesised, which only showed the conventional N phase, the first reported experimental observations of a polar ferroelectric nematic phase were not made until 2017 for RM734[6,7] and DIO[8]. The lowest temperature nematic phase in RM734 was investigated by Chen et al.[9] who reported the bulk polar response characteristic of a ferroelectric phase and therefore proposed that this polar nematic phase was the $N_F$ phase. It was later found in RM734 that there were in fact two nematic phases below the conventional nematic phase[10,11]. The intermediate, narrow temperature range nematic phase was assigned as the splay nematic phase, $N_S$, due to the detection of a significant pretransitional behaviour which manifested strong splay orientational fluctuations leading to its periodic structure[12–14]. It was suggested that a modulated phase would fill the space more efficiently and that this could be achieved in a one-dimensional, *single splay*, or a two-dimensional, *double splay*, structure. These were formed by alternating regions in which molecules pointed upwards or downwards, arising in locally polar domains separated by splayed directors[15]. Interestingly, the same phase sequence, I-N-$N_S$-$N_F$, was also seen in the other archetypical ferroelectric nematogen, DIO[8].

In contrast to the structure of the conventional N phase, in the $N_F$ phase there is a spontaneous alignment of the molecular dipoles, and the inversion symmetry associated with the N phase is lost, i.e. n ≠ -n, resulting in a polar phase. The ferroelectric nematogens reported to date are mostly low molar mass compounds exhibiting a large longitudinal dipole moment and generally have lateral substituents such as alkyloxy groups or fluorine atoms[16,17]. These structural features result in the formation of molecules with non-centrosymmetric shapes and strong attractive forces, and in this respect are in general consistent with computer simulations reported by Berardi et al. that predicted the $N_F$ phase for pear-shaped particles[18]. A molecular model to account for the formation of the $N_F$ phase has been proposed by Madhusudana[19] in which the molecules are represented as cylindrical rods with alternating regions of positive and negative charge. The structure is described by a longitudinal surface charge density wave. The model suggests that the parallel alignment of the molecules is enhanced by minimising the amplitude of the charge density wave at either end of the molecule, and by increasing the amplitude of the charge density wave at the centre of the molecule.

Most of the research to date on these materials has focused structural features taken from RM734[6,7], DIO[8] or UUQU-4-N[20]. The structure of RM734 has been the most extensively varied and these modifications have included varying the terminal and lateral substituents, such as using different alkyl or alkyloxy chains[3,7,21–28], changing the fluorination patterns[29–33], the introduction of biphenyl moieties[34] and different terminal groups[31,35–37]. Among these, a more limited number of materials were found to exhibit a direct $N_F$-I transition[21–23,29,32]. A few examples of liquid crystal polymers have also been investigated, including main-chain[38,39] and side-chain polymers[40], but these examples are sporadic and highlight the difficulty in designing ferroelectric nematogens whose structure deviates from the archetypal compounds defined earlier. It is evident that to realise the application potential of the $N_F$ phase[41–46], we must develop a better understanding of the structure-property relationships responsible for driving its formation in order to underpin the rational design of new materials having targeted properties. Moreover, the investigation of the $N_F$ phase has already led to the discovery of other nematic phases, including the antiferroelectric $N_X$ phase[26,32,47,48] also referred to as the $SmZ_A$ phase[49,50] or the $N_S$ phase[13], and the polar heliconical $N_{T_BF}$ phase[51–53] as well as polar smectic phases[2,54–64], also making these materials interesting from a fundamental point of view.

We have previously reported on the behaviour of ferroelectric nematogens based on the structure of RM734, in which a lateral alkyloxy chain attached to either the central benzoyl ring, NT3.*m* and NT3F.*m* series[22], or the terminal benzoyl ring, **5**-*m* and **6**-*m* series[21], was varied in length, Fig. 1a. These series highlighted the possibility of significantly altering the overall shape of the molecules without disrupting the $N_F$ phase and showed that the conventional nematic phase was much more sensitive to changes in shape. This was further explored in later studies via addition, substitution or removal of methoxy groups with other groups[23,27,33]. Here we expand this knowledge combining structural features presented earlier, in a comprehensive manner in order to test the tolerance of the $N_F$ phase to withstand these structural modifications. We introduce 70 ferroelectric nematogens to the current library of around 250 compounds. While there are some examples of pure compounds[38,65,66] and multi-component mixtures[55,67,68] which can exhibit the $N_F$ phase at room temperature after some degree of supercooling, to date there has only been one example of a mesogen with a transition to the $N_F$ phase below 30 °C, UUQU-4N[20]. Therefore the modifications outlined in this work aimed to drive the transition temperatures of the $N_F$ phase to values around room temperature. A common feature in all the reported series is a lateral alkyloxy chain, attached to either the central or terminal aromatic ring which is varied in length from two to seven or eight methylene units. This further explores the effects of large lateral groups on the formation of the $N_F$ phase but also is expected to lower the transition temperatures of these materials towards room temperature. The structures of the reported compounds are shown in Fig. 1b, in each of the acronyms, *m*, designates the number of carbon atoms in the lateral alkyloxy chain, while H, F, Me or OMe refers to hydrogen, fluorine, methyl or methoxy substituents, respectively, on the aromatic rings. Series A.*m*, B.*m*, C.*m* and D.*m* are based on the structures shown in Fig. 1a with a replacement of the terminal MeO group by a Me group. Series E.*m*, F.*m*, G.*m* and H.*m* again have a replacement of the terminal MeO with a Me group, but in addition have an additional lateral MeO group added to one of the aromatic rings. Series I.*m*, J.*m*, K.*m* and L.*m* are directly based on the structures in Fig. 1a but with the additional lateral MeO group added to one of the aromatic rings.

## Results

The transitional properties of the twelve series are summarised in Tables S30–41 in the ESI. Of the 86 ferroelectric nematogens reported, the great majority form the $N_F$ phase directly upon cooling from the isotropic; only three exhibit a preceding nematic phase (A.1, A.2 and C.1). We have previously reported the phase behaviour of several of the shortest homologues: A.1[27], B.1[27], C.1[27], D.1[26,27], E.1[27], F.1[27], G.1[27], H.1[27], I.1[23], I.2[23], J.1[23], J.2[23].

Phases were initially assigned based on observations made using polarized optical microscopy. Specifically, nematic phases formed characteristic schlieren textures containing both two- and four-point defects and which flashed when subjected to mechanical stress, Fig. 2a, or a uniform planar texture in cells treated for planar alignment. For those $N_F$ phases formed from the N phase, at $T_{N_FN}$ domain walls emerged, which separated regions of differing birefringence, giving the so-called banded texture, which has become characteristic for the $N_F$ phase, Fig. 2c–f. These domains are thought to mark areas where the direction of the polarisation differs from one another. In instances where the $N_F$ phase was formed directly from the isotropic phase, the banded texture tended to develop from the coalescence of spherical droplets which emerged at $T_{N_FI}$. Longer homologues with $T_{N_FI}$ around room temperature did not exhibit the typical banded texture associated with the $N_F$ phase (e.g. Fig. 2g, h), presumably because of the high viscosity of the material at low temperatures, though a clear phase transition was observed.

Due to the absence of characteristic optical textures for the $N_F$ phase in these samples, the phase assignment was confirmed by the

**Fig. 1 | The molecular structures of previously reported compounds and the structures of compounds reported in this manuscript. a** The molecular structures of the NT3.*m*, NT3F.*m*, **5**-*m* and **6**-*m* series, which were previously reported[21,22]

and **b** The molecular structures of the A.*m* (*m* = 1–8), B.*m* (*m* = 1–8), C.*m* (*m* = 1–7), D.*m* (*m* = 1–7), E.*m* (*m* = 1–7), F.*m* (*m* = 1–7), G.*m* (*m* = 1–7), H.*m* (*m* = 1–7), I.*m* (*m* = 1–7), J.*m* (*m* = 1–7), K.*m* (*m* = 1–7) and L.*m* (*m* = 1–7) series.

construction of a binary phase diagram using mixtures of a representative example, K.6, and the archetypal ferroelectric nematogen RM734, Fig. 3. Complete miscibility of the two materials was observed over the whole composition range investigated, indicating the same phase behaviour in both components. Only the mixture containing 80 mol% RM734 exhibited the conventional nematic phase above the $N_F$ phase, and all others showed a single mesophase. Textural observations of the characteristic schlieren texture (N) and banded texture ($N_F$) confirmed the assignments of both phases. The values of $T_{N_F I}$ decreased linearly as the mole fraction of RM734 decreased and the virtual transition temperature extrapolated for the pure compound was 6 °C, in good agreement with the measured temperature of 10 °C. Based on this assignment, it is reasonable to assume that the other compounds with similar viscous mesophase textures also exhibit the $N_F$ phase.

Representative DSC traces for D.7, G.5 and L.3 are shown in Fig. 4. All show a large endotherm during the initial heat cycle associated with the melting point, and on cooling a weaker exothermic peak is observed corresponding to the $N_F$-I transition. Interestingly, all three of the samples shown are stable to room temperature and even with subsequent reheating do not recrystallise. Of the compounds reported here some 52 are stable on cooling at or below room temperature and perhaps more notably 19 have values of $T_{N_F I}$ below 30 °C. This is quite a surprising number considering the rarity of single-component ferroelectric nematogens with a transition to the $N_F$ phase at room temperature with UUQU-4N as a notable example[20], and this observation

will be discussed in more detail later. However, it must be noted that most of the $N_F$ transitions are monotropic in nature and in some cases these transitions are considerably below the melting point. The values measured for $\Delta S_{N_F I}/R$ are wholly consistent with this phase assignment and in good agreement with those reported previously[21–23,27]. Although most of these room temperature $N_F$ compounds remained stable while held at room temperature, there were some which exhibited a glass transition on further cooling towards 0 °C as shown in Fig. SI4 and Tables SI30–41.

The dielectric studies, Fig. 5, support the microscopic assignment of the $N_F$ phase. The dielectric permittivity is measured as a function of applied frequency and temperature, showing a sudden increase upon entering the polar nematic phase. A strong relaxation mode is observed, with a characteristic frequency that varies from a few Hz to kHz, depending on the material. The mode frequency decreases with decreasing temperature. Although the exact nature of the active relaxation process remains to be determined, this response is clearly related to the collective reorientation of the dipoles. The materials were also examined for their nonlinear optical response. In the $N_F$ phase, a strong second-harmonic generation, arising from the non-centrosymmetric structure of the phase, was detected. The spontaneous polarization values were typical for this class of materials (~5 μC/cm²) and were slightly higher for materials with a fluorine substituent at terminal phenyl ring, which is understandable given their somewhat larger dipole moments, Fig. SI5. Although the spontaneous polarization was

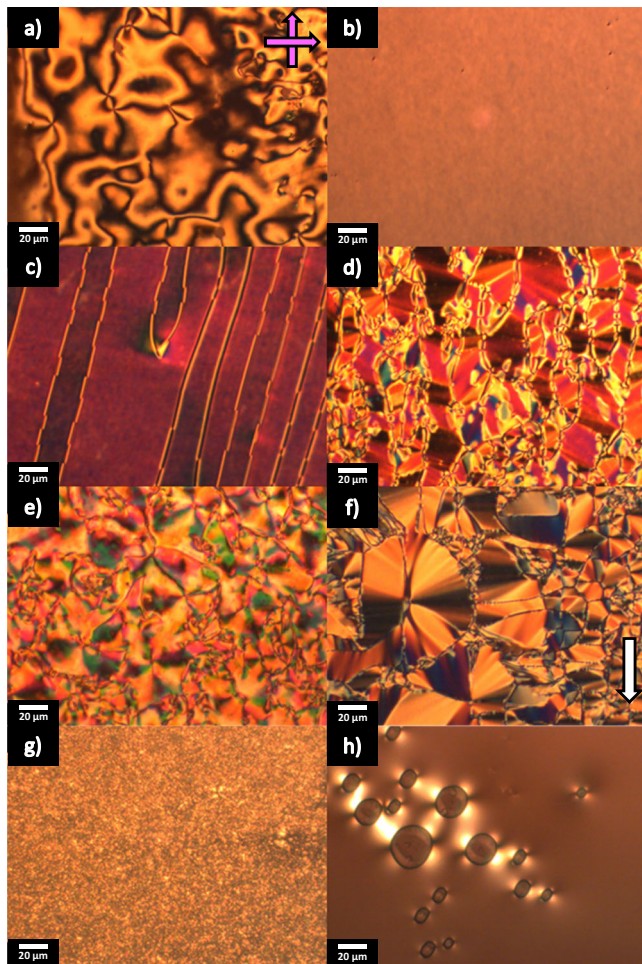

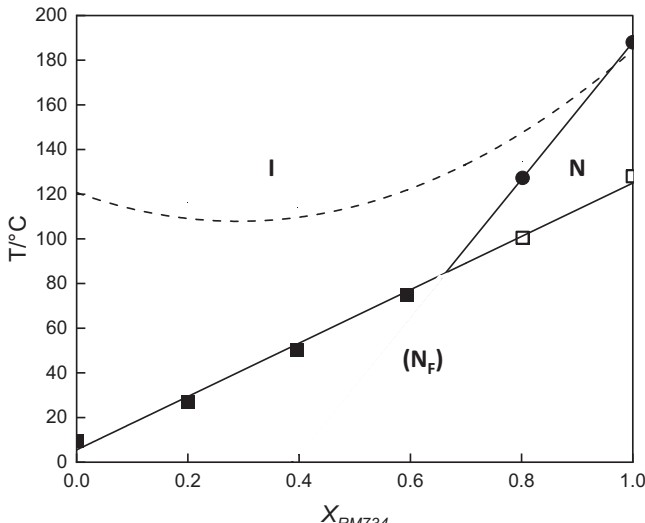

**Fig. 3 | Phase diagram for mixtures of K.6 and RM734 plotted as a function of the mole fraction of RM734, $X_{RM734}$.** The filled circles represent $T_{NI}$, the filled squares $T_{N_FI}$, the empty squares $T_{N_FN}$ and the broken line connects the melting points. The solid lines indicate trend lines drawn for $T_{NI}$ and $T_{N_FN/I}$.

**Fig. 2 | Representative optical textures of selected compounds from the reported series. a** The schlieren texture of the N phase for A.2 ($T = 104\,°C$); **b** Uniform texture of the N phase in a planar aligned cell for A.2 ($T = 104\,°C$); **c** Banded texture of the $N_F$ phase for A.2 ($T = 98\,°C$); **d** Banded texture of the $N_F$ phase for B.2 ($T = 104\,°C$): **e** Banded texture of the $N_F$ phase for D.4 ($T = 87\,°C$): **f** Banded texture of the $N_F$ phase in a planar aligned cell for L.3 ($T = 48\,°C$): **g** Uncharacteristic texture of the $N_F$ phase for L.7 ($T = 23\,°C$) and **h** homeotropic texture of the $N_F$ phase for K.6 ($T = 22\,°C$). For all POM images, the polariser and analyser axes are shown by the pink arrows. The white arrow shows the direction of the planar alignment of the cell.

similar, the materials differed significantly in their switching time, specifically in the delay at which the repolarization current peak appears upon reversal of the electric field. Selected materials, with a long lateral chain, $m = 7$, were studied in order to determine how the changing position of this chain influenced the material's rotation viscosity (Fig. 5). Near room temperature, the delay of polarization switching peak was on the order of tens of milliseconds for compound H.7 (for electric field magnitude $2\,V\,\mu m^{-1}$), while for D.7, C.7 and B.7 it was at least one order of magnitude smaller. The delay can be expressed as $\tau = \gamma/(EPs)$, where $\gamma$ is the rotational viscosity. This relation allowed the viscosity associated with polarization reorientation to be estimated and in all the measured materials, it increased strongly on cooling, following an Arrhenius dependence. The viscosity was found to be strongly dependent on molecular structure: the highest in material H.7 containing two lateral chains (with $m = 7$ chain in the middle phenyl ring and a methoxy group in terminal ring) and F atom *ortho* to the nitro group. Removal of a short methoxy lateral substituent (D.7) reduced the viscosity by an order of magnitude. Moving the long lateral chain from the central to

the terminal ring (B.7) reduces the viscosity further. Removing the F atom substituent (C.7) also reduced the viscosity, however the effect was weaker, Fig. SI5. In general, for all materials with long lateral substituents the rotational viscosity was much higher, than for the parent RM734 nematogen[9]. Interestingly, the rotational viscosity corresponds to the dielectric response, so the larger the viscosity, the lower the frequency of the dielectric mode, Fig. 5.

## Discussion

Of the twelve series reported here, only two materials exhibit the conventional nematic phase, A.1, A.2 and C.1. As the number of methylene units in lateral chain increases the N phase is destabilised and the extent of this destabilization is more prominent than that of the $N_F$ phase such that direct I-$N_F$ transitions are observed for the other members of those series. The N phase is also extinguished with any additional lateral substituents, such as the addition of a fluorine atom or methoxy group, which is completely in line with the conventional theory of the nematic phase for which a decrease in shape anisotropy leads to a general destabilisation of the phase[69].

The dependence of $T_{N_FN/I}$ against the lateral alkyl chain length, $m$, for all the reported series and literature compounds of similar structures: A.$m$, B.$m$, C.$m$, D.$m$, E.$m$, F.$m$, G.$m$, H.$m$, I.$m$, J.$m$, K.$m$ and L.$m$, 5·$m$[21], 6·$m$[21], NT3.$m$[22] and NT3F.$m$[22] are compared in Fig. 6. All of the series show a similar temperature trend with increasing lateral chain length: namely the values of $T_{N_FN/I}$ initially decrease but converge to reach a limiting value when $m = 6$, completely in agreement with our previous reports for other materials with lateral chains[21,22]. On top of this general trend, $T_{N_F-}$ differs depending on the nature of the substituents on the molecular core. First, we consider the effect of adding a F atom *ortho* to the terminal nitro group, Fig. 6a: the reported fluorinated series consistently show an increase in the values of $T_{N_FN/I}$ when directly compared to their non-fluorinated counterpart. A model proposed by Madhusudana suggests that parallel, ferroelectric alignment of molecules will be promoted by rod-like structures with low amplitudes of surface charge density at each terminus[19,21]. Such a model can rationalise our observation by considering the *ortho* F atom acts to spread electronic charge across the terminal aromatic ring, lowering the amplitude of the surface charge density wave at this end of the molecule and thus a stabilisation of $N_F$ would be expected. This is indeed the case, although the extent of this increase varies

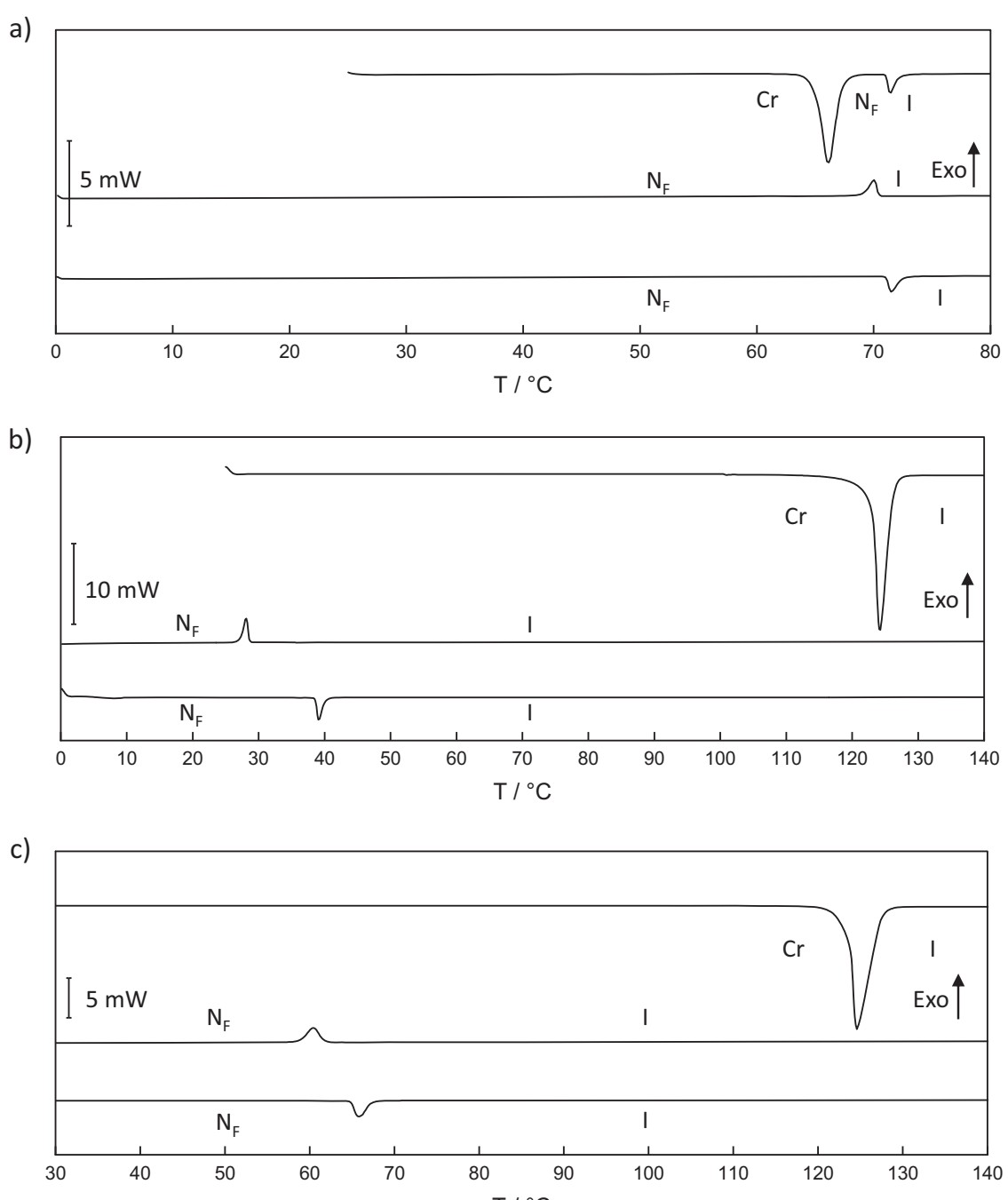

**Fig. 4 | Representative DSC traces measured for selected ferroelectric nematogens.** DSC traces obtained for **a** D.7, **b** G.5 and **c** L.3. On each trace is shown the (top) initial heat, (middle) cool and (bottom) reheat cycles. The direction of the exothermic peaks in these traces is marked by the arrow labelled Exo.

depending on the rest of molecular structure. The largest difference in $T_{N_F N/I}$ is observed between series G and H, around 12 °C, whereas the smallest is between series J and I with the latter on average only 3 °C higher. It is also possible that the additional F atom changes the orientation of the aromatic rings, rotating the terminal phenyl ring with respect to the central ring, and hence the shape of the molecule, but also the direction of the vector for the dipole moment due to the negative charge associated with the F atom. These steric and electronic effects, and the interplay between them on altering the chemical structure, will also contribute to the efficiency of molecular packing into a ferroelectric arrangement.

We can also consider the effect of the position of the lateral alkyloxy chain on the occurrence of the $N_F$ phase, Fig. 6b. In general,

when the alkyloxy chain is in the middle aromatic ring the $N_F$ transition temperatures are higher and this is seen particularly when comparing the NT3.$m$[22] series and the **5**-$m$[21] series from literature where the $N_F$ transition temperatures are around 30 °C higher for the former. The difference in the behaviour depending on the position of the original lateral alkyloxy chain can again be justified by the model proposed by Madhusudana. Changing the position of the lateral alkyloxy chain changes the electron distribution across the aromatic rings and so will influence the amplitude of the charge density wave. When the alkyloxy chain is attached to the terminal aromatic ring, the amplitude at this end of the molecule will be higher due to the electron donating effect of this functionality. Conversely, when the alkyloxy chain is in the middle ring the amplitude will be now larger in the middle ring and

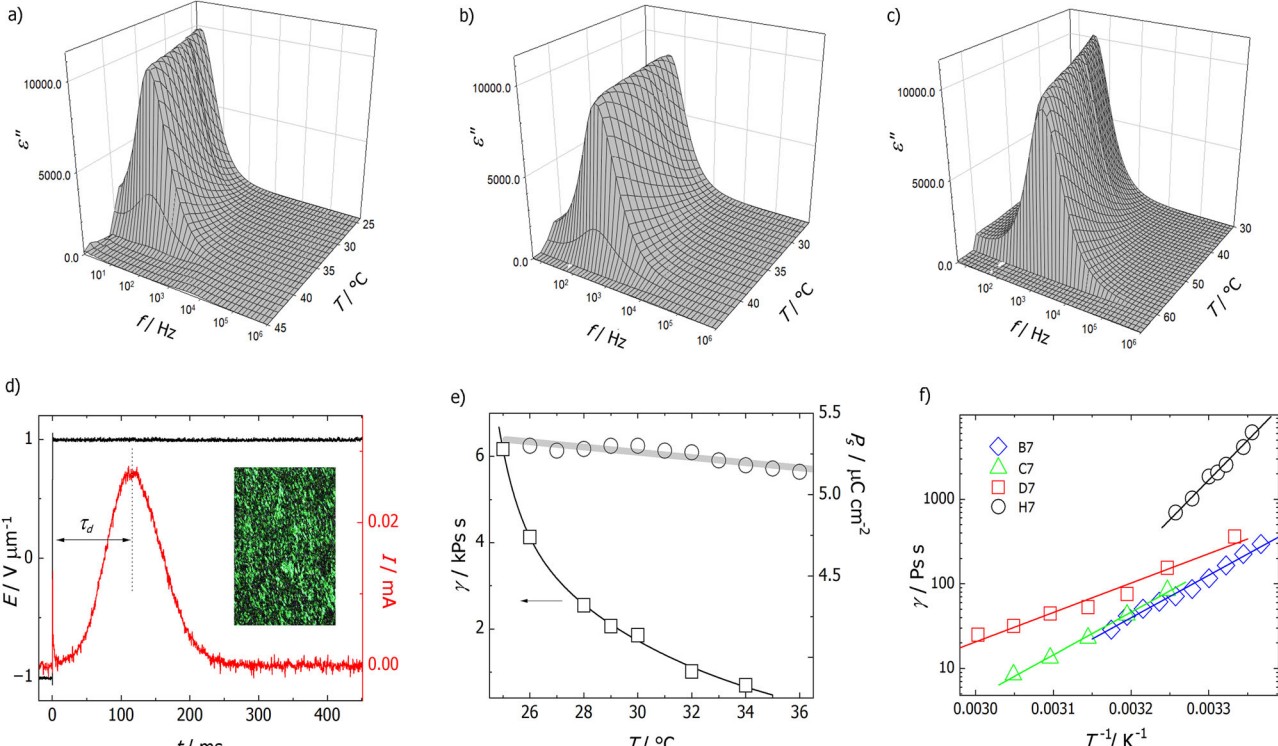

**Fig. 5 | The dielectric, polarisation and rotational viscosity measurements of representative compounds with long lateral chains.** The frequency-temperature dependence of the imaginary part of the dielectric permittivity for compounds **a** H.7, **b** B.7 and **c** C.7. **d** Current peak (red line) observed during polarization reversal under application of square-wave electric field (black line) for compound H.7 at 24 °C. Time delay of the peak maximum, $\tau_d$, was used for determination of rotational viscosity. In the inset, image taken with SHG microscope for $N_F$ phase of H.7 compound at 25 °C, illumination of the sample with IR ($\lambda = 1064$ nm) radiation causes emission of green (SH) light, proving ferroelectric character of the phase. **e** Rotational viscosity (squares) and spontaneous electric polarization (circles) measured vs. temperature in $N_F$ phase for compound H.7. **f** Arrhenius plots of rotational viscosity measured vs. temperature for compounds B.7 (blue diamonds), C.7 (green triangles), D.7 (red squares) and H.7 (black circles). Activation energy determined from the slope of fitted linear dependence (lines) changes from 0.7 eV for D.7 to 1.9 eV for H.7.

relatively lower at the terminus, Fig. 7, and so will favour the parallel packing of molecules. This effect was previously described by Tufaha et al.[22]. This effect is not uniform, however and both the K and L series exhibit lower values of $T_{N_FI}$ than the I and J series, respectively. Presumably, this is related to the number of electron-donating groups and steric bulk associated with the substituents on the aromatic rings, but it would be expected the electronic profile would be similar to the other series and so the physical significance of this observation is not yet clear.

Next, we consider the effect of changing number of lateral groups in the molecule with the addition of a methoxy group to either the left or central aromatic rings depending on the position of the pre-existing lateral alkyloxy chain, Fig. 6c. For the four series with the alkyloxy chain *m* on the left-hand aromatic ring, namely the A, **5**-*m*, B, and **6**-*m* series, $T_{N_FN/I}$ decreases by around 20 °C upon addition of the second lateral substituent on the middle ring to give the E, I, F and J series, respectively. The decrease observed for the four series which have their original lateral alkyloxy chain on the middle aromatic ring and the additional MeO group laterally substituted to the terminal aromatic ring, namely series C, NT3.*m*, D and NT3F.*m* is almost twice as large: $T_{N_FI}$ is reduced by on average 39 °C when compared to the doubly laterally-substituted series G, K, H and L, respectively. Such a decrease is entirely expected given the reduction in shape anisotropy with the additional methoxy group, Fig. 7. When the second MeO group is added to the terminal aromatic ring this will increase the amplitude of the charge density wave at that end of the molecule which will reduce the ability of the molecules to pack in a parallel manner as well as decreasing shape anisotropy. However, adding MeO instead to the middle aromatic ring decreases the shape anisotropy but its electronic

contribution at the centre of the molecule—acting to increase the charge density in the centre of the molecule—balances the effect of the lateral alkyloxy chain on the terminus, and hence the cumulative effect is a smaller decrease in $T_{N_FI}$.

Next, we study the effect of the nature of the terminal group has on the phase behaviour and compare methoxy- and methyl-terminated counterparts, Fig. 6d. Again, there is a clear dependence on the position of the lateral alkyloxy chain. In general, when the alkyloxy chain is on the terminal aromatic ring such as in series A, E, and B, the $N_F$ phase is stabilised by 9 °C, 2 °C, and 3 °C, respectively, when the terminal methyl group is replaced by a methoxy group (giving series **5**-*m*, I, and **6**-*m*, respectively). $T_{N_FI}$ of series F, however, instead shows a small decrease of 2 °C, upon making this change (to series J). The former observation is somewhat surprising as this exchange should cause an unfavourable increase in amplitude of the surface charge density wave at the terminus. Indeed, all of the series with the alkyloxy chain in the middle show a much larger decrease of around 12 °C in the values of $T_{N_FN/I}$ when the terminal methyl group is replaced by a methoxy group, in line with what would be expected based on the model proposed by Madhusudana[19]. This difference in behaviour between those compounds with the original alkyloxy chain on the terminal ring and those with it on the central ring can be somewhat rationalised again by the fine balance of electronic effects: presumably the strong electron donating effect of the lateral alkyloxy group already present in the terminal ring means that the relative effect of changing the donating effect of a methyl group, $\sigma_p = -0.17$, for a methoxy group, $\sigma_p = -0.27$[70], is reduced compared to when the alkyloxy group is present in the middle ring. It would therefore seem that the combination of a terminal methoxy and lateral alkyloxy chain in the terminal ring more

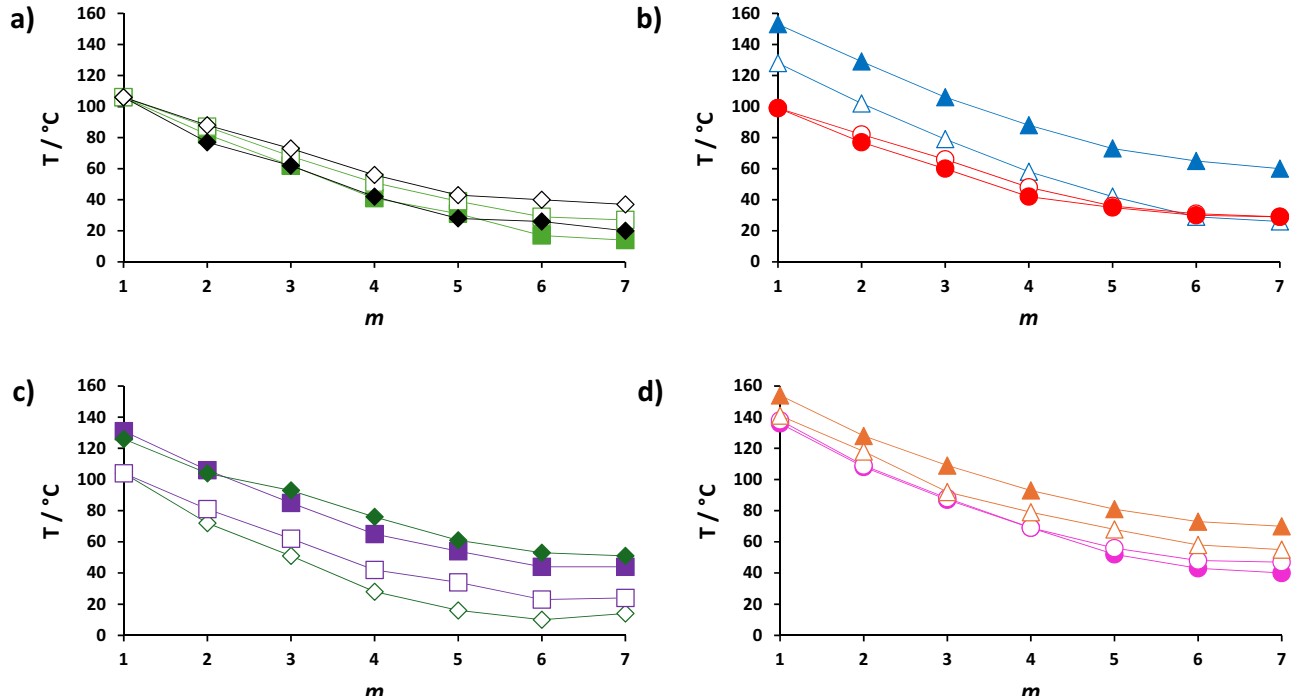

**Fig. 6 | Dependence of the ferroelectric nematic transition temperatures** ($T_{N_F N/I}$) **on the number of carbon atoms in the lateral alkyloxy chain,** *m,* **for the reported series. a** The E.*m* series is represented by filled light green squares, F.*m* series by empty light green squares, G.*m* series by filled black diamonds and H.*m* series by empty black diamonds; **b** The C.*m* series is represented by filled blue triangles, A.*m* series by empty blue triangles, L.*m* series by red filled circles and J.*m* series by empty red circles; **c** The **5**-*m* series is represented by filled purple squares, I.*m* series by empty purple squares, NT3.*m* series by filled green diamonds and K.*m* series by empty green diamonds; **d** The D.*m* series is represented by filled orange triangles, NT3F.*m* series by empty orange triangles, B.*m* series by filled pink circles and **6**-*m* series by empty pink circles.

favourably promotes the packing of molecules in a parallel manner compared to a methyl and alkyloxy chain, highlighting the very complex nature of the interactions which drive the ferroelectric nematic phase. This is somewhat surprising since we have previously reported that increasing size of the terminal group in this ring decreases the stability of the $N_F$ phase[26]. For example, increasing the terminal chain from a methyl to an ethyl group saw $T_{N_F I}$ decrease by 18 °C and a similar effect was also observed by Mandle et al.[7] for other compounds of the RM734-type. Another method which has been utilized in literature to control the charge distribution within the aromatic ring system of ferroelectric nematogens has been the incorporation of aromatic heterocycles[36,71]. Interestingly, while being very monotropic and requiring a supercool of over 150 °C, a pyridine-based $N_F$ material reported by Mandle exhibits a transition to the $N_F$ phase at 45 °C, a value more in line with the compounds reported in this work, than with RM734 itself.

What should be noted is the number of compounds with $N_F$-I transitions at or below 30 °C, with only 1 previous example currently reported for a pure compound, UUQU-4N[20]. Indeed, even the widely used room temperature $N_F$ mixture FNLC 919[68] cools into the $N_F$ at a temperature >30 °C. Our compounds maintained the $N_F$ phase even with long lateral alkyloxy chains attached to the mesogenic unit[21,22]. This highlights that the controllability of $T_{N_F I}$ by variation in the nature and size of the lateral substituents. Using **5**-6[21] from the literature as an example: $T_{N_F I}$ = 44 °C, which is quite close to room temperature. However, by adding an additional MeO group to the middle ring we could expect the transition temperature to decrease based on our previous reports[23], and this is indeed what is observed: $T_{N_F I}$ of I.6 is 23 °C, a room temperature transition to the $N_F$ phase. This additional methoxy group acts to decrease the shape anisotropy and consequently the molecular packing efficiency, thus lowering $T_{N_F I}$. A reliable method to decrease $T_{N_F N/I}$ towards room temperature is extension of

the lateral alkyloxy chain. A combination of multiple lateral groups and extension of this chain can therefore be utilised to achieve room temperature transitions of materials within the RM734 structure space. However, room temperature $N_F$-I transitions were not exclusively seen just for compounds with multiple lateral alkyloxy groups. Three members of series A.*m*, with *m* = 6–8, have $T_{N_F I}$ below 30 °C. Compound **5**-6 which is reported in literature[21] has a value of $T_{N_F I}$ = 44 °C. If its terminal MeO group is replaced by a Me group to give its counterpart A.6, $T_{N_F I}$ decreases to 29 °C. Despite the latter appearing to be more electronically favourable for driving the parallel packing described by Madhusudana's model[19], this comparison highlights that there are still limitations in our understanding of the interplay between shape and electronic effects in driving $N_F$ phase formation.

However, we can still utilise the knowledge accrued from our molecular library and this model to make strategic simultaneous molecular changes and optimise $T_{N_F N/I}$. For example, adding a MeO group to the terminal ring of NT3.6[22] causes $T_{N_F I}$ to decrease from 53 to 10 °C in K.6, but by adding a F atom *ortho* to the $NO_2$ terminus the $N_F$ phase can be stabilised again such that L.6 has a $T_{N_F I}$ of 30 °C. In this work we report 19 compounds which exhibit $N_F$ phase transitions at 30 °C or below. Our next target is therefore reduction of the melting points in these materials through further structural modifications to attain highly desirable enantiotropic, room-temperature ferroelectric nematogens suitable which would be suitable for use in devices. The changes observed in the stability of the $N_F$ phase translate into the rotational viscosity associated with polarization switching dynamics. Molecules containing long lateral substituents exhibit an increased cross-sectional area, which enhances steric hindrance and strengthens intermolecular interactions. As a consequence, the rotation of molecules is perturbed, leading to a significant increase in the rotational viscosity, γ. Removing even a single lateral substituent or moving the longer substituent from central to terminal ring markedly decreases

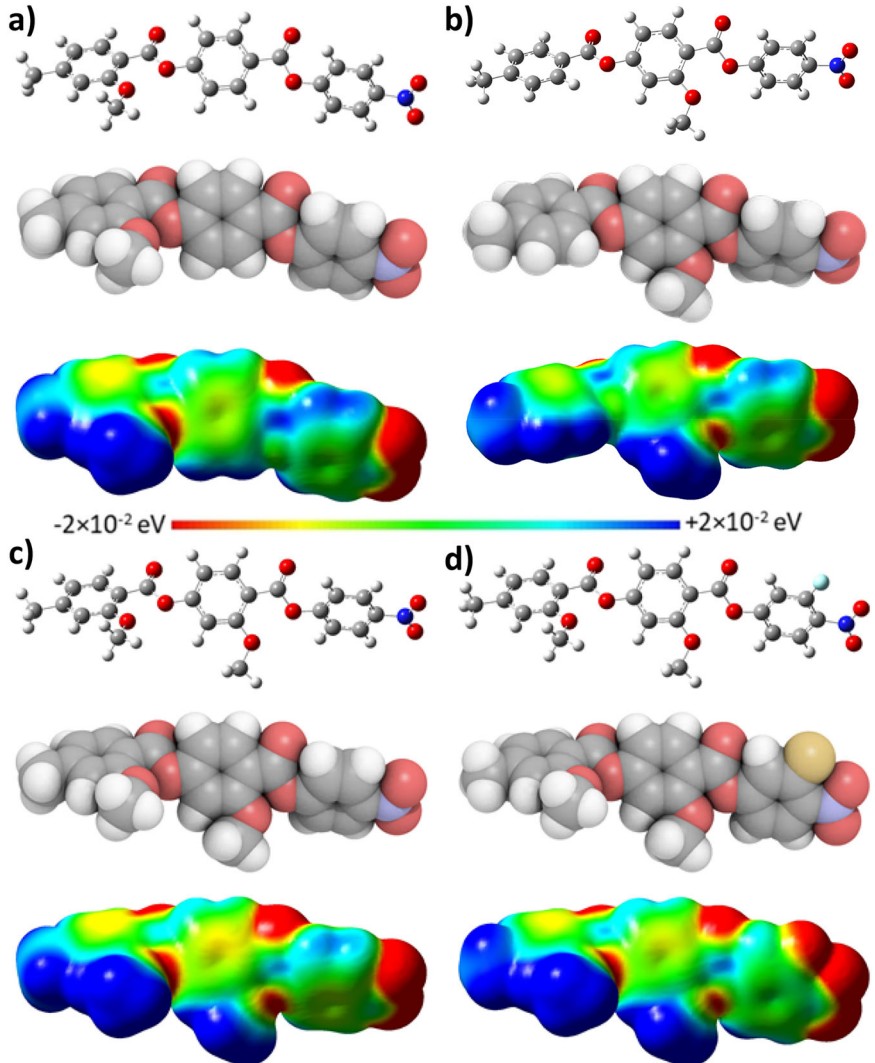

**Fig. 7 | Molecular modelling of compounds with a methoxy lateral group.** Ball-and-stick models (top), space-filling models (middle) and electrostatic potential surfaces (bottom), with red regions marking areas of negative charge and blue regions marking areas of positive charge, of: **a** A.1, **b** C.1, **c** E.1 and **d** F.1 calculated at the B3LYP/6-31, level of theory.

steric effect, and thus the viscosity can be reduced by an order of magnitude. By adjusting viscosity through molecular design, one can achieve either fast-response materials or stable polar state. Such control broadens the range of potential applications, from rapid electro-optic devices to robust, long-lived ferroelectric elements.

We report twelve series of compounds which all exhibit the $N_F$ phase based on the RM734 structural template. Structural modifications included: changing the nature of the terminal group between methyl and methoxy moieties; the position of the lateral alkyloxy chain along the mesogenic core; the number of lateral alkyloxy groups present; the degree of fluorination; and the length of the lateral alkyloxy chain. We have used a combination of these changes to further identify trends in phase behaviour, particularly in terms of the stability of the $N_F$ phase, and these were related to the computational model proposed by Madhusudana[19]. All of the presented series show a similar temperature trend with increasing lateral chain length, the values of $T_{N_FN/I}$ initially decrease but converge to reach a limiting value when $m = 6$. When an additional lateral methoxy group is added $T_{N_FN/I}$ decreases, and this decrease is larger when the original alkyloxy chain is attached to the middle ring of the molecule compared to when it is on the terminal ring. When a fluorine is added *ortho* to the terminal nitro group, the $N_F$ phase is stabilised compared to the non-fluorinated counterpart series,

but the extent of this increase varies depending on the rest of the molecular structure. When the lateral alkyloxy group is on the terminal ring, replacing the terminal methyl group with a methoxy group increases the stability of the $N_F$ phase. However, when the chain is instead on the middle ring, replacing the terminal methyl group with a methoxy group instead decreases the stability of the $N_F$ phase. Of the 70 compounds which have been added to the $N_F$ structure space, 19 exhibit the phase transition to the $N_F$ phase below 30 °C with only 1 previous example currently reported for a pure compound[20]. This shows that by carefully using specific structural changes it may be possible to tailor the transition temperatures of these materials. Both, the phase stability and the viscosity of ferroelectric materials, which governs polarization switching dynamics, can be tuned by molecular design, allowing control over the balance between switching speed and long-term retention of the polarized state.

## Methods

### Synthesis

The synthetic route used to prepare the compounds is shown in Figs. 8–10. A detailed description of the preparation of all the intermediates and final products, including full structural characterisation, is provided in the Supplementary Information.

**Fig. 8 | Synthetic scheme for the A.*m* and B.*m* series.** The compounds were synthesised using a five-step reaction with the final step being a Steglich esterification.

**Fig. 9 | Synthetic scheme for the C.*m*, D.*m*, G.*m*, H.*m*, K.*m* and L.*m* series.** The compounds were synthesised using a six-step reaction with the final step being a Steglich esterification.

## Reagents

All reagents and solvents that were purchased commercially were obtained from Sigma Aldrich, Fisher Scientific, TCI or Fluorochem and these were used without further purification.

## Thin layer chromatography

The progress of the reactions was monitored using thin layer chromatography with the appropriate solvent system. These were aluminium-backed plates with a coating of Merck Kieselgel 60 F254

**Fig. 10 | Synthetic scheme for the E.*m*, F.*m*, I.*m*, J.*m* series.** The compounds were synthesised using a five-step reaction with the final step being a Steglich esterification.

silica which were purchased from Merck KGaA. UV light (254 nm) was used to visualise the spots on the plate.

## Column chromatography
Separations carried out with normal phase column chromatography, used silica gel grade 60 Å, with a particle size of 40−63 μm, purchased from Fluorochem and using an appropriate solvent system.

## Structure characterisation
All synthesised final products and intermediates were characterised using $^1$H NMR, $^{19}$F NMR and $^{13}$C NMR and infrared spectroscopies. The infrared spectra were recorded on a Perkin Elmer Spectrum Two FTIR spectrometer with an ATR diamond cell. The NMR spectra were recorded on a 400 MHz Bruker Avance III HD NMR spectrometer

## Purity analysis
The purity of the final products was confirmed using high-resolution mass spectrometry, using a Waters XEVO G2 QTof mass spectrometer by Dr. Jayne McCaskill at the University of Aberdeen, or elemental analysis. The C, H, N microanalysis was carried out by either: the Sheffield Analytical and Scientific Services Elemental Microanalysis Service at the University of Sheffield using an Elementar Vario MICRO Cube or the Elemental Analysis Services Team at the London Metropolitan University using a ThermoFlash 2000.

## Optical studies
Initial characterisation of the mesophases was performed by polarised optical microscopy, using a Zeiss AxioImager A2m equipped with a Linkam THMS600 hot stage. Slides treated for homotropic alignment were purchased from AWAT with a thickness of 1.7 μm.

## Differential scanning calorimetry
The thermal behaviour of the materials was studied by differential scanning calorimetry performed using a Mettler Toledo DSC1 or DSC3 differential scanning calorimeter equipped with TSO 801RO sample

robots and calibrated using indium and zinc standards. The heating and cooling rate used was 10 K min$^{-1}$, with a 3 min isotherm between either heating or cooling. All samples were measured under a nitrogen atmosphere. For each sample, two aliquots were measured, and the data listed are the average of the two sets. The transition temperatures and associated enthalpy changes reported were extracted from the heating traces unless otherwise noted.

## Molecular modelling
Optimisation of the molecular structures was carried out using quantum mechanical DFT calculations processed by Gaussian09 software[72] at the B3LYP/6-31 G(d) level. Visualisations of electronic surfaces and ball-and-stick models were generated from the optimised geometries using the GaussView 5 software. The electronic surfaces were found with the cubegen utility in GaussView by generating a total density cube using a SCF density matrix and course grid, overlayed by an ESP surface map. Visualisations of the space-filling models were produced post-optimisation using the QuteMol package[73].

## Dielectric spectroscopy
The complex dielectric permittivity, $\varepsilon^*$, was studied using a Solatron 1260 impedance analyser. Measurements were conducted in the 1 Hz−1 MHz frequency ($f$) range, with the probe voltage of 20 mV, and it was checked by optical observations that such a voltage is below the Fredericks transition threshold. The material was placed in 20 μm-thick glass cells with gold electrodes and no surfactant layers. Lack of a surfactant layer resulted in the random configuration of the director in the LC phases; microscopic observations of optical textures suggested a dominant planar orientation without preferable direction of the long molecular axis. The relaxation frequency, $f_r$, and dielectric strength of the mode, $\Delta\varepsilon$, were evaluated by fitting the complex dielectric permittivity to the Cole−Cole formula: $\varepsilon - \varepsilon_\infty = \sum \frac{\Delta\varepsilon}{\left(1+\frac{if}{f_r}\right)^{1-\alpha}} + i\left(\frac{\delta}{2\pi\varepsilon_0 f}\right)$ where $\varepsilon_\infty$

is the high frequency dielectric constant, $\alpha$ is the distribution parameter of the mode and $\delta$ is the low frequency conductivity, respectively.

## Spontaneous polarization and rotational viscosity

Spontaneous polarization, $P_s$, was measured using 20 μm-thick cells with gold electrodes and no surfactant layers, a triangular-wave voltage of frequency 1–10 Hz was applied. The polarization value was calculated by analysing the current flow through a resistor (1K-10K Ohm) connected in series with the cell under testing. A Siglent SDG2042X arbitrary waveform generator, FLC A200 amplifier and Siglent SDS2000X Plus oscilloscope were used. The rotational viscosity was calculated from the delay time, $\tau_d$, between the current peak and applied square wave voltage step. The frequency of voltage applied was 1 Hz, and a series of measurements was performed as a function of electric field magnitude. Using the relation $\tau_d = \gamma/EP_s$ the rotational viscosity, $\gamma$, was obtained.

## Second harmonic generation (SHG) microscopy

The SHG microsopy images were obtained with a setup based on a solid-state laser EKSPLA NL202, with 9 ns pulses at a 10 Hz repetition rate and max. Two megajoule pulse energy at $\lambda = 1064$ nm. The infrared beam incident onto a cells with planar anchoring resulted in generation of green light.

## Data availability

The authors declare that the data supporting the findings of this study are available within the paper, the supplementary information files and the source data file. The unprocessed data for the work is also available from the corresponding author upon request. Source data are provided with this paper.

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

## Acknowledgements

The authors would like to highlight the vital contribution to this work made by Professor Corrie Imrie, who passed away suddenly in January 2025. C.T.I. and J.M.D.S. acknowledge the financial support of the Engineering and Physical Sciences Research Council [EP/V048775/1]. D.P. and E.G. acknowledge the financial support of the National Science Centre (Poland) [2024/53/B/ST5/03275].

## Author contributions

E.C., N.T. and G.S. synthesised and performed structural characterisation of the materials, undertook thermal analysis and modelled the materials using Gaussian; E.C., N.T., G.S., R.W., D.P. and E.G. undertook optical studies; N.T., D.P., and E.G. performed dielectric measurements; D.P. and E.G. performed polarisation and SHG measurements; C.T.I., J.M.D.S., D.P., E.G. proposed the project and acquired funding; E.C., N.T., G.S., and R.W. wrote the original manuscript; all authors participated in the analysis and discussion of the project and provided feedback on the manuscript.

## Competing interests

The authors declare no competing interests.
