## [Transparent Peer Review file · Nature Communications]

Investigating room temperature ferroelectric nematogens and their structure-property relationships

Corresponding Author: Dr Ewan Cruickshank

Version 0:

Reviewer comments:

Reviewer #1

(Remarks to the Author)

Tufaha and colleagues investigated a variety of substituted aromatic esters for developing ferroelectric nematogens in their study. These compounds were devised by modifying the end group, the position of the lateral alkyloxy chain, the number of lateral alkyloxy groups, the degree of fluorination, and the length of the lateral alkyloxy chain. While the connection between chemical structure and phase behaviour has been thoroughly explored, I recommend that this work be submitted to a liquid crystal journal that specialises in more specific research within the field.

After reviewing the referenced literature, I believe that the strategies for modifying molecular structures discussed in this study have already been extensively examined by R. Mandle, the authors themselves, and others, rendering the work lacking in originality. The conclusions drawn are intuitive, lacking in-depth analysis and precise calculations of dipole and electron density. Throughout the paper, the main focus remains on the phase behaviour of these molecules, without delving further into the relationships between structure and properties. Overall, this work does not offer any novel insights or reveal intriguing material properties.

Reviewer #2

(Remarks to the Author)

[Editorial Note: See end of file]

A pdf file "Initial Review Comments" has been uploaded

Reviewer #3

(Remarks to the Author)

In the manuscript entitled "Investigating room temperature ferroelectric nematogens and their structure-property relationships", Tufaha et al. report on the synthesis and characterization of a large number of compounds exhibiting the ferroelectric nematic (NF) phase, many of which were stable down to room temperature. This constitutes an important step forward, not only in the understanding of the mechanisms driving the formation of the NF phase, but also in the search for enantiotropic room temperature ferroelectric nematogens suitable for applications. In fact, to date only one pure compound exhibited the NF phase around room temperature. The work presented in this manuscript is therefore of high value and should undoubtedly be considered for publication in Nature Communications. However, despite the undeniable quality of this work, there are some aspects in the manuscript that need to be polished and I, thus, call for a major revision. In this regard, there are two main points that the authors should address:

1. There is only indirect evidence that these compounds show the NF phase (textures in POM, enthalpy peaks in DSC, miscibility studies and large dielectric response). I do not doubt it but, in order to rigorously prove it, the authors should perform either spontaneous polarization or second-harmonic generation measurements (ideally both), at least in some

selected compounds. This is basic in the field of ferroelectrics.

2. One of the most remarkable outcomes of this work is the identification of the structural changes that promote the NF phase and lower transition temperatures. For example, it is stated that "a reliable method to decrease" NF transition temperatures is "extension of the lateral alkyloxy chain". Nonetheless, the authors do not assess at any point in the manuscript what consequences these structural changes may have on the physical properties of the materials. This is a relevant question for devices and applications, because it is not enough to have a room temperature NF phase; it should also have properties tuned for specific applications. For instance, it would be reasonable to assume that the aforementioned side-chains would increase the rotational viscosity of the liquid crystal, thus slowing down switching times. Fluorination is also used as a strategy in this paper, which will affect the material properties as well. What happens with other properties like birefringence, polarization, nonlinear optical response and so on? These also depend on the molecular structure, of course. Ideally, the authors should study some of these properties in selected compounds and look for trends. But assessing it and discussing it in some depth should be considered as the bare minimum.

Other minor comments:

1. I identified a couple of typos. In the abstract, lines 16-17, it should read "detailed analysis" instead of "detail analysis". In the introduction, line 37, "provided those interactions" instead of "providing those interactions".

2. In the introduction, the text in lines 38-50 is somewhat inaccurate and should be rephrased. It gives the impression that the assignment of the phase immediately below the N phase on cooling in RM734 to the splay nematic (NS) phase was incorrect. Although we now know that RM734 presents the NF phase, there is a narrow NS phase between the N and NF phases, i.e., the phase sequence on cooling is N-NS-NF. Apart from the initial studies that showed it (as well as some subsequent ones), this was proven by precision calorimetry in 2024: <https://doi.org/10.1103/PhysRevE.110.014703>.

3. In Fig. 3 the authors should add a sketch of the polarizer/analyzer configuration somewhere.

4. With regards to the dielectric measurements, in line 140 authors say that they use 9.7 micron-thick cells, while in line 204 they say 5 microns.

Version 1:

Reviewer comments:

Reviewer #2

(Remarks to the Author)

This is reviewer #2.

Comments for Author

My original review tended to be on the verbose side and so the follow-up comments provided here on the response received from the authors will be brief.

I am generally quite happy with the replies from the authors to my comments found the original review.

The authors have responded constructively, debugged the MS and made it more accessible to the reader. Most importantly, the authors have added additional information on a variety of issues that were raised by myself and the other reviewers.

In particular, a discussion of the issue of viscosity (a physical property of significant consequence here) has now been included in the revised manuscript. For example, Figure 6 now shows us a snapshot about viscosity in addition to the earlier description limited to dielectric permittivity.

Hopefully, future work on structure and property relationships can decouple tradeoffs between phase identity, and their respective temperature and viscosity. As noted previously in the original review, this accomplishment would be a real "breakthrough" distinct from iterative progress.

These issues go back a long time for "simple nematic" small molecules (and including oligomers and LC polymers containing them). FYI, here are just a few representative references I happen to know from the 20th century, examining the influence of lateral substituents in phenylbenzyloxybenzoates:

J. Org. Chem., 37, 3707-3712 (1972)

J. fur Prakt. Chem., 321, 47-64 (1979)

Liq. Cryst., 10, 623-624 (1991)

I include these references here as I wanted to share them with the authors and did not have them readily available during the original review.

By exploiting these materials and performing more comprehensive analysis still more improved examples can be created inspired by what was learned from their progenitors. These issues are now emphasized in the closing of the Conclusion

Section of the revised manuscript.

Reviewer #3

(Remarks to the Author)

The authors have satisfactorily assessed all of my comments. I recommend the article for publication in Nature Communications.

Reviewer #1 (Remarks to the Author):

Tufaha and colleagues investigated a variety of substituted aromatic esters for developing ferroelectric nematogens in their study. These compounds were devised by modifying the end group, the position of the lateral alkyloxy chain, the number of lateral alkyloxy groups, the degree of fluorination, and the length of the lateral alkyloxy chain. While the connection between chemical structure and phase behaviour has been thoroughly explored, I recommend that this work be submitted to a liquid crystal journal that specialises in more specific research within the field.

After reviewing the referenced literature, I believe that the strategies for modifying molecular structures discussed in this study have already been extensively examined by R. Mandle, the authors themselves, and others, rendering the work lacking in originality. The conclusions drawn are intuitive, lacking in-depth analysis and precise calculations of dipole and electron density. Throughout the paper, the main focus remains on the phase behaviour of these molecules, without delving further into the relationships between structure and properties. Overall, this work does not offer any novel insights or reveal intriguing material properties.

Answer: We thank the referee for their comments, and although some modifications of the RM734 compound have indeed been provided by R. Mandle and other groups, our studies are first in which the molecular structure has been systematically varied to such an extent. In the revised version, we have incorporated new experimental data on both the polarization of the materials and the viscosity associated with polarization switching and linked them to molecular structure. These results provide a more detailed structure–property analysis of these materials. By quantifying how specific structural modifications influence polar behavior and switching dynamics, the work now goes beyond describing phase behavior and offers clearer insights into how molecular design affects functional material properties. We feel the work here has important insights on structural design which will be greatly beneficial to delivering the potential of these materials.

Reviewer #2 (Remarks to the Author):

Page 2 Abstract

“... allowing for a detailed analysis of the structure-property relationships” What is the journal policy about use of Arabic numerals vs. numbers “written out in full”? The journal will decide this grammar issue. Examples: characterization of twelve. All seventy of the new compounds. Some fifty-two could be supercooled ... and other similar instances in the manuscript.

Answer: We thank the referee for their comment but have left the numbers as Arabic for now and will change them if required by the journal’s policy, however we have corrected the typographical error.

“Only one pure compound to date has exhibited the NF phase at this low temperature, UUQU-4N...” This blanket statement above about UUQU-4N is found in the abstract (and thus also without immediate elaboration or citation). Are there no other single pure compounds in the literature in addition to UUQU-4N with such a low temperature NF phase (or at least at or near ambient)? Perhaps some literature claims are incorrect (or actually do not apply to single component systems or fail for some other reason?). Single pure oligomers should be considered (even if they may otherwise ultimately fail somehow). Please elaborate on the claim made in the abstract about the uniqueness of the low

temperature phase of UUQU-4N. Some simple Google searches turned up some possibilities for published at or near roomtemperature NF liquid crystals. What about: <https://doi.org/10.1039/D5TC02738A> Room-temperature ferroelectric nematic liquid crystals: the influence of substituents on properties(cite and discuss this recent publication which may postdate submission of this MS)DOI:10.1002/adv.202414317 Ferroelectric Nematic Liquid Crystals Showing High Birefringence(mixtures vs. single component?) currently citation 61 <https://arxiv.org/pdf/2309.14161> Room-temperature ferroelectric nematic liquid crystal showing a large and divergent density(mixture?) not currently cited? <https://doi.org/10.1080/02678292.2024.2345214A> ferroelectric nematic liquid crystal vitrified at room temperature "... phase behaviour leading to an NF phase from 63°C down to room temperature." not currently cited? <https://www.tandfonline.com/doi/full/10.1080/02678292.2021.1921867>Ferroelectric nematic phase at and below room temperature "We report a liquid crystal 1 that shows ferroelectricity at room temperature" currently citation 18 <https://doi.org/10.1021/jacs.1c09594>. How Far Can We Push the Rigid Oligomers/Polymers toward Ferroelectric Nematic Liquid Crystals? currently citation 36 <https://doi.org/10.1021/acs.macromol.1c00864> Polar Liquid Crystalline Polymers Bearing Mesogenic Side Chains with Large Dipole Moment currently citation 38 This is not a definitive search. The reviewer does not have access to some full articles. Some of these articles may not be relevant. In any case, it is worthwhile to validate the claim. Thus far it has certainly been a challenge to identify single component materials which have an NF phase at or near ambient temperature. The search for such materials is worthy but ultimately is this not the pursuit of the idiomatic sacred cow in a liquid crystal context? Mixtures are commonly (if not invariably utilized) in lieu of unitary composition to attain the requisite wide variety of function required for liquid crystal applications. In this regard, is the examination of some mixtures of these new compounds a worthy investment of time and effort? It appears mixtures with RM734 have been discussed here but only in the context of assisting phase identification (Figure 4 of the manuscript). Please elaborate.

Answer: We thank the referee for their comments, and they are correct that a number of manuscripts report the N_F phase at room temperature although they tend to have higher melting temperatures but can be supercooled. The phraseology here was a little clumsy on our part and so did not capture that UUQU-4N has a transition at room temperature itself rather than being able to be stabilized at room temperature and so this has been amended in the abstract. Bar UUQU-4N, which is reference 18 mentioned by the referee, all the other N_F mixtures or compounds have a transition above 30 degrees. Here we report 52 new compounds with low temperature NF phases. To address this a section has been added to the introduction about some of the current materials which can be cooled to room temperature and highlighting how we are reporting examples with the transition at room temperature rather than higher temperature NF phases which can be supercooled. Like in the abstract the phraseology has been made specific about the criteria being used. We agree that the next step for these materials is a wide-ranging investigation of mixtures to optimize the phase transition temperatures and to control the melting points. However, we believe this would go beyond the scope of this work, as this would require a significant amount of time and optimization to test all the feasible combinations and to test how they change the properties. However, this is something that will be considered as future work to this manuscript. By having a component near or at room temperature one could reduce the need for large numbers of components to make mixtures and so more careful control can be established for optimization. Critically to test a wide range of mixtures with different combinations would be a project by itself which deserves time and proper consideration.

A dioxane ring (at least in the context of DIO) is not a lateral group since it is not attached to the side of the molecule. In DIO the 5-alkyl-1,3-dioxane ring attached at the 2-position is a terminal group.

Answer: We thank the referee for spotting this error and we have now corrected it.

Page 4

Seventy new ferroelectric nematogens in the manuscript ... a substantial number added to the existing about two hundred and fifty. Common feature is a lateral alkoxy chain. The discussion provided about the structure variations is well managed given the scope of structures found in the literature. This manuscript follows from previous papers and previous strategy from this group (including references 19, 20 21,25, 31, 32 ... others?).

Answer: We thank the referee for the positive endorsement of our work.

Page 5

Consider adding "previously reported" to the caption of Figure 1. Figure 2 shows the "reported compounds" in this new manuscript which number $2 \times 8 + 10 \times 7 = 86$ compounds (of which seventy are ferroelectric)? Would it be helpful at this stage to explain up front the differences between Figure 1 and Figure 2? For example, the introduction of the 4-methyl group instead of a methoxy group is quite comprehensive in A.m through H.m while E.m through L.m have two lateral alkoxy groups not seen previously in Figure 1.

Answer: We thank the referee for their comment and have added that remark to the figure caption of figure 1, and an additional section has been added specifying how the structures in Figure 2 vary compared to the literal structures shown in Figure 1.

Page 6 – Page 8

The manuscript is devoted and limited to the modification of the system $C_6H_5-COO-C_6H_4-COO-C_6H_4-NO_2$ involving replacement of various hydrogen atoms with a selected limited number of substituents Me, OMe, OR (longer linear alkoxy), and F in different combination, number and locality. These few variables already deliver a huge matrix of options which are being systematically addressed (and obviously numerous other such modifications are conceivable). The synthesis schemes pages 6 through page 8 are clear and the experimental methods pages 8 through page 9 are clear. The synthesis steps are not described in much detail (and has already been done in earlier related publications). The variability in esterification yields are worthy of some comment. There appears to be a change in nomenclature/style between Figure 1 and Figure 2 and the style found in Scheme 1 through Scheme 3. In Figure 1 and Figure 2 " $OCmH_{2m+1}$ " appears for the lateral alkoxy groups while in Scheme 1 through Scheme 3 " Om " is used. The latter style seems unnecessary and is somewhat unorthodox. Please comment on style consistency (and even as it pertains to style in earlier publications cited).

Answer: We thank the referee for their comment and noticing this difference, we have therefore homogenized the schemes and figures so that the lateral chain is labelled in the same way to help readers. The esterification yields will be discussed in answer to the ESI question later. We also thank the referee for their comments on the synthetic scheme and methods.

Page 8- Page 9 Various Measurements / Modeling. This subsection is fine except the absence of any polarisation measurements is surprising. This group has reported polarization measurements in some previous publications (ref 20, ref 21). The reviewer has no reason to doubt the existing phase assignments made in the absence of polarization measurements. Nonetheless, polarization

measurements constitute a main feature in most discussions about all ferroelectric substances. Please comment on the absence of at least some representative polarization measurements.

Answer: We thank the referee for their comment and the spontaneous electric polarization for several representative materials have been added to evaluate the influence of the molecular modifications. In addition, SHG measurements were included to strengthen the phase identification.

Page 9 Results

Eighty-six ferroelectric nematogens reported and twelve of them are previously reported. Only three nematogens are reported to possess a standard nematic phase in addition to the ferronematic phase. Is there any benefit to having this adjacent nematic phase (such as for alignment purposes)? Please elaborate about the impact of an adjacent nematic phase on the utility of the nematic ferroelectric phase.

Answer: We thank the referee for the question. The presence of nematic phase in temperature range above the N_F phase indeed helps in achieving uniform alignment of the phase in cells with homogenous anchoring. However the effect was not significant; also for materials with direct Iso-NF transition, upon sufficiently slow cooling (in cells with parallel rubbing of polymer layers on both substrates) a sufficiently good alignment can be obtained.

Page 10 – Page 12

The high viscosity of (some or all?) materials at low temperatures is noted in the context of issues with texture assignment. This appears to be the only mention of viscosity issues of these materials in the entire manuscript. How pervasive is this viscosity issue? Do the “low temperatures” mentioned here include room temperature? Are there any performance consequences associated with the high viscosity (e.g., switching dynamics)? Some applications of NF materials may not be impacted by high viscosity (or could even benefit from it?). So, while the desired NF phase exists at lower temperatures in many of these new materials, are there some performance trade-offs due to high viscosity near ambient temperatures in these new compounds which should be considered? Phase assignments reported on the high viscosity materials. required the use of a binary mixture. Are mixtures also going to be necessary in some potential application(s)? These issues are overlooked so far and require elaboration (and, if the issues really are troublesome, then solutions are required). Oligomers and polymers of *p*-hydroxybenzoic acids have been widely studied. The viscosity of such polyesters is a critical processing parameter. Perhaps this is a place to learn something about the influence of substituents on phase behavior and viscosity. It is not clear whether or not any existing information is relevant to ferronematic materials, but it seems worthwhile to look at since significant effort has gone into the understanding and optimization of LC polyesters. The reviewer did some quick searching about relationships between functionalization of these polymers and viscosity but did not turn anything very relevant up. Sorry. Such information must be available.

Answer: We thank the referee for their comment and have undertaken measurements as well as adding extra detail to our discussion. The viscosity of the studied materials is indeed an important issue, as it strongly influences the polarization-switching dynamics. Therefore, measurements were performed for several selected compounds and correlated with their molecular structures. In general, for mesogens with long lateral chains the rotational viscosity was found to be very large (and strongly increasing with lowering temperature). Consequently, despite the high polarization values, the switching times remained in the millisecond range for most of the studied materials with long lateral chains. The rotational viscosity could be manipulated to some extent, mainly through position and number of

lateral substituents in mesogenic core. A full paragraph discussing the rotational viscosity of the studied materials has been added to the main text.

Page 13

Why is reference 18 cited in relation to some forthcoming discussion? Based on the discussion on this page about the glass transition the reviewer anticipated seeing an example of a DSC with this feature. While information about this may be included somewhere in Tables S28-S29 for the benefit of readers unfamiliar with this issue can you also show an actual DSC trace with this feature?

Answer: We thank the referee for spotting this error and have corrected the location of the reference. We have also added an example of a DSC as suggested (Figure S11) which shows a glass transition to help readers with understanding how this presents itself on the trace.

Page 14 - Page 15

Dielectric permittivity data is provided to support assignment of the NF phase (and, as noted, in spite of associated controversy about anomalous large values). As already noted, there is a conspicuous absence of any polarization measurement in this manuscript that often is a critical component of numerous other papers on this topic. What is lost (if anything) by not having some polarization data to examine at this stage?

Answer: We thank the referee for this observation and so the full frequency-temperature dependence of dielectric loss is now presented in the new Figure 6, along with spontaneous polarization vs. temperature for selected compounds in the same figure and in the Supporting Materials, Figure S12. An SHG microscopy image is presented (inset in new Figure 6) as additional proof of ferroelectric character of NF phase.

Page 16

Are citations available to support the statement “conventional theory of the nematic phase for which a decrease in shape anisotropy leads to a general destabilisation of the phase”? If this statement is accurate and can be applied broadly (including ferronematics) some additional discussion would be helpful. Please provide background on this. In Figure 7 “Dependence of the transition temperatures on the number of carbon atoms in the lateral alkoxy chain” is presented. The materials have multiple phase changes. Specifically, which phase change(s) are plotted? Can this be succinctly indicated in the figure or in the caption?

Answer: We thank the referee for noting there was no reference listed for this comment and have added one to Imrie et al. ‘The preparation and properties of low molar mass liquid crystals possessing lateral alkyl chains’ which makes reference to the effect of shape on TNI. We also thank the referee for spotting the mistake in the figure caption which did not expressly say the transitions were for the NF phase and this has now been changed.

Page 17 – Page 18

Some of the structure variations of substituents on RM734-like substances found in research to date is discussed. Absent from the discussion are reported substitutions of heterocycles for the benzene rings. Shuffling some (hetero)atoms in the rings is an equally legitimate method to optimize ferroelectric behavior (existence, magnitude, temperature range, etc.) as the same activity on the rings. For example, see: <https://doi.org/10.1039/D4TC02851A> Sulfur-based ferroelectric nematic liquid crystals (thiophene) <https://doi.org/10.1080/02678292.2022.2145380> Supramolecular ferroelectric nematic

materials (pyridine) doi.org/10.1002/cplu.202300726 The Emergence of a Polar Nematic Phase: A Chemist's Insight into the Ferroelectric Nematic Phase Currently citation 14 (compound 83, pyridine)

Answer: We thank the referee for pointing out that heteroaromatics had been omitted and we have added a small paragraph referring to them in the discussion with the aforementioned references.

Page 19

In Figure 8, where is the arrow indicating the positive direction of the calculated dipole moment mentioned in the caption?

Answer: We thank the referee for spotting this error and have removed the comment from the caption.

Page 20

Discussion is presented in this section about the comparison of methoxy and methyl terminal groups. The key difference between the methoxy group and a methyl group is an oxygen atom (two "large" atoms vs. one). Simple structure changes have less than simple outcomes.

Answer: We thank the referee for their comment and completely agree with the sentiment.

Page 21

Using 5-619 from the literature. The 5-619 is not clear to the naïve reader (but may be to the authors familiar with this nomenclature). Should this read 5-619 ... italicized consistent with Figure 1? "... there are still limitations in our understanding of the interplay between shape and electronic effects in driving NF phase formation" AGREED "Our next target is therefore reduction of the melting points in these materials through further structural modifications to attain highly desirable enantiotropic, room-temperature ferroelectric nematogens suitable which would be suitable for use in devices." GOOD LUCK Enantiotropic NF behaviour is rare or absent amongst the new compounds in the manuscript. Does the basic three ring diester system examined here provide adequate "bandwidth" for delivering the desired enantiotropic NF phase (including at or near ambient temperature)?

Answer: We thank the referee for spotting the missing word and have added this. We thank the referee for this discussion, it does seem likely that the ester functionality may need to be replaced by, for example, a CF₂O linkage which is found in UUQU-4N which appears to lower the melting point compared to an ester linking group and so this may be the initial types of modifications made to see if we could get closer to an enantiotropic transition. Although some examples with this linkage also have not been NF and so again careful control will be needed to maintain the polar phases but to at the same time lower the melting point. The nomenclature is for the compound in the article referenced but this has been edited to make the line more reader friendly.

Page 22 – Conclusions

"This shows that by carefully using specific structural changes it may be possible to tailor the transition temperatures of these materials. Indeed, if the melting points these materials could be lowered it does raise the exciting possibility that a room temperature enantiotropic NF material is very much feasible." AGREED as a concept It appears likely that further improvements in phase behaviour will be found in the RM-734 class of compounds. But here is a strategy consideration: perhaps there is some currently unidentified system which has more "dynamic range" and is also easier to manipulate to deliver lower temperature ferroelectric phases along with other acceptable physical properties. So, while there is

value in digging in deeply within a specific format (e.g., RM734 type substances) there is also the risk that it may not end up being the most expedient or effective format to modify. What say yes?

Answer: We thank the referee for their comment and agree entirely that there may well be an even more optimal structure for obtaining the properties outlined. The goal of doing such a wide ranging study was to try and test the limits of existing structures known to exhibit the phase which will allow lessons to be taken for optimization of those structures when they are identified. We do appreciate the referee supporting our concept.

Comments on supplementary information

Generally, the supplementary information section is meticulously and thoroughly prepared and presented. This is much appreciated. The entire SI technical data section was NOT examined with a fine-tooth comb.

Answer: We thank the referee for their positive comment on our work.

SI Page 8

Colourless oil. Yield: 4.32 g, 91 %. RF: 0.33 (100% dichloromethane)

Answer: We thank the referee for spotting this typographical error and have corrected it.

Table SI3, SI5 and others

In this study DCC is used most often for Steglich esterifications. The yields for various DCC esterification reactions vary quite a bit (here and elsewhere) and are sometimes quite low. Is there some reason for this? Is there any rhyme or reason (intrinsic to the reaction itself or variability in the workup/isolation)? The yields are what they are ... just wondering about this in case there may be some remedy to suggest. There is not much in the way of synthesis methodology discussed in the main text but hopefully this issue can be discussed in the main text or otherwise in the supplementary information.

Answer: We thank the referee for their positive comments on the presentation of the ESI. With regards to the esterification reactions, the differences in yields are likely related to a combination of solubility of the starting materials, the purification process and the solubility of the products due to the different type and position of substituents on the molecular backbone. Specifically, the change in chain length, for example, will change significantly the solubility of the product in the final recrystallization. Some of these reactions also suffered from side-reactions such as transesterification, which led to both an inherent reduction in the yield of desired product and a need for increased purification steps, which led to further reductions.

Artwork / Nomenclature for Synthesis Schemes 1-4 Some of the artwork/nomenclature utilized is unconventional. For example, in Scheme 1 compound (1) has a substituent at the 2-position which is "Om" where "m" is the number of carbon atoms in the chain. Typically, in usual descriptive parlance, this transformation would be indicated with the new 2-substituent being OC_mH_{2m+1}. So, this is not wrong ... just unconventional and different from style used elsewhere. This issue was mentioned earlier about the Synthesis Schemes in the main text.

Answer: We thank the referee for their observation of this and have changed it to match elsewhere with C_mH_{2m+1}.

Table SI9.1, Table SI9.3 The esterification reagent is now EDC. Is there some rationale to use one carbodiimide instead of the other (here and elsewhere)? Again, while there is not much synthesis discussion in the manuscript any insights like this could be valuable.

Answer: We thank the referee for their comment and the reason for this difference was purely down to the compounds being synthesized by different authors and the availability of different reagents at the time of synthesis. EDC is generally more reactive than DCC and so was likely used to try and increase the yield of the intermediate being formed. We chose to use DCC for the majority of final products because this method was one modified from the original which aimed to avoid issues with transesterification and proved successful.

Additional Data

It would be very helpful to add general structures at the top of Tables SI28 – SI39 so the reader can be spared seeking this info elsewhere. There are eighty-six compounds found in Tables SI28 – SI39 (of which seventy are new?). Is it possible to provide some indication (perhaps a citation) for the compounds which have been previously published? This additional bookkeeping would help clarify progress in this class of compounds. On page nine of the manuscript the following twelve entries have been identified as already reported: A.1,25 B.1,25 C.1,25 D.1,24,25 E.1,25 F.1,25 G.1,25 H.1,25 I.1, 21 I.2,21 J.1,21 J.2.21. The inclusion of this information in the Tables would be helpful. In a number of places in the manuscript there are mentions of “x of this flavor” and “y of this flavor”, etc. The categories and populations are of interest. Can they somehow be summarized in the existing tabulation?

Answer: We thank the referee for their comment and have added the general structures of each series to their corresponding table. We have also added a note to the tables to mark previously published materials as suggested and thank the referee for pointing out this oversight.

Summary

This manuscript is from a joint project at the University of Aberdeen and the University of Warsaw. This manuscript follows from directions and successful outcomes which are described in a number of significant previous publications. Generally speaking, a strength of the manuscript and its supporting information is that they are meticulously prepared. A strength of the manuscript is the sheer number of individual new substances described and the discovery that a significant fraction is ferronematic. A current weakness of the manuscript is the lack of additional characterization data (e.g., polarization, x-ray). Some earlier publications by this joint project include polarization measurements. Comprehensive physical measurements on such a large number of new compounds are not practical and are not required, but at least some representative measurements would be instructive and supportive. Of course, an argument might be made that they may not. Please reply. The current manuscript focusses heavily on the behavior of individual compounds. Knowledge about the physical properties of single compounds is valuable but a practical solution ultimately lies in the formulation of mixtures. Should this mixture approach be applied in parallel to the development of individual compounds? What is the best set of physical properties to measure? These are some generic questions. A current and primary weakness of the manuscript is the paucity of discussion about the origin and/or impact of the high viscosity observed in some of these new NF materials. The reviewer is concerned this feature may impede ultimate practical applications of this class of materials. It appears that structure modifications implemented to lower the NF phase temperature range have been successful, but they may well also bring about an increase (or at least permit revelation) of a high viscosity régime near room temperature. Better understanding of the structure/viscosity relationship will turn iterative progress into a breakthrough which will pay off here and could likely pay off for other materials classes as well. Earlier, the understanding and development of simple nematic liquid crystals went through a protracted period dealing with viscosity issues. It is no surprise the same issue is relevant with ferronematics. The methods and mechanisms leading to improvement in the simple nematics may or

may not apply to the ferronematics. Please address the viscosity issue. The reviewer looks forward to hearing back from the authors. Does some existing information assist in solving about the viscosity issue? What appropriate additional characterization will assist to obtain a solution to the viscosity issue?

Answer: We thank the referee for their positive comments. With regards to characterization we have now added the spontaneous electric polarization for several representative materials to evaluate the influence of molecular modifications. In addition, SHG measurements were included to strengthen the phase identification. Indeed, the molecular modifications significantly changed the rotational viscosity of the compounds. In general, the viscosity was found to be very high (and resulted in polarization response in the time domain of milliseconds) for compounds with long lateral alkoxy chains. As was mentioned in one of the earlier responses, extra details have been added with regards to the viscosity of the materials and how that has influenced some of the other properties. In terms of changing the viscosity of the compounds, using mixtures for example, would be a technique that could be utilized to try and help reduce the viscosity while maintaining the transition temperature as described for example by Pramanik et al, Liq. Cryst. 40 (2013), 149-158 and we showed to some extent with the binary mixtures with RM734 in the manuscript which were in general less viscous by visual inspection of the microscope slides. But as suggested previously we believe that these mixtures are worthy of their own piece of work.

Reviewer #3 (Remarks to the Author):

In the manuscript entitled "Investigating room temperature ferroelectric nematogens and their structure-property relationships", Tufaha et al. report on the synthesis and characterization of a large number of compounds exhibiting the ferroelectric nematic (NF) phase, many of which were stable down to room temperature. This constitutes an important step forward, not only in the understanding of the mechanisms driving the formation of the NF phase, but also in the search for enantiotropic room temperature ferroelectric nematogens suitable for applications. In fact, to date only one pure compound exhibited the NF phase around room temperature. The work presented in this manuscript is therefore of high value and should undoubtedly be considered for publication in Nature Communications. However, despite the undeniable quality of this work, there are some aspects in the manuscript that need to be polished and I, thus, call for a major revision. In this regard, there are two main points that the authors should address:

Answer: Thank you very much for the favorable comments.

1. There is only indirect evidence that these compounds show the NF phase (textures in POM, enthalpy peaks in DSC, miscibility studies and large dielectric response). I do not doubt it but, in order to rigorously prove it, the authors should perform either spontaneous polarization or second-harmonic generation measurements (ideally both), at least in some selected compounds. This is basic in the field of ferroelectrics.

Answer: We thank the referee for their comment and In response, measurements of both spontaneous polarization (Ps) and second-harmonic generation (SHG) have been added for selected compounds. These provide direct evidence for the presence of the NF phase.

2. One of the most remarkable outcomes of this work is the identification of the structural changes that promote the NF phase and lower transition temperatures. For example, it is stated that "a reliable

method to decrease" NF transition temperatures is "extension of the lateral alkyloxy chain". Nonetheless, the authors do not assess at any point in the manuscript what consequences these structural changes may have on the physical properties of the materials. This is a relevant question for devices and applications, because it is not enough to have a room temperature NF phase; it should also have properties tuned for specific applications. For instance, it would be reasonable to assume that the aforementioned side-chains would increase the rotational viscosity of the liquid crystal, thus slowing down switching times. Fluorination is also used as a strategy in this paper, which will affect the material properties as well. What happens with other properties like birefringence, polarization, nonlinear optical response and so on? These also depend on the molecular structure, of course. Ideally, the authors should study some of these properties in selected compounds and look for trends. But assessing it and discussing it in some depth should be considered as the bare minimum.

Answer: We thank the reviewer for the suggestions. Indeed, the molecular modifications significantly change the rotational viscosity as the referee suggests. In general, the viscosity was very high (and resulted in polarization response in the time domain of milliseconds) for compounds with long lateral alkoxy chains. The additional data related to polarization value and switching dynamics as well as rotational viscosity were provided – new Figure 6 and Figure SI2. We have also written in further detail about the properties of these materials to give additional depth to the discussion.

Other minor comments:

1. I identified a couple of typos. In the abstract, lines 16-17, it should read "detailed analysis" instead of "detail analysis". In the introduction, line 37, "provided those interactions" instead of "providing those interactions".

Answer: We thank the referee for spotting these typographical errors and have corrected them.

2. In the introduction, the text in lines 38-50 is somewhat inaccurate and should be rephrased. It gives the impression that the assignment of the phase immediately below the N phase on cooling in RM734 to the splay nematic (NS) phase was incorrect. Although we now know that RM734 presents the NF phase, there is a narrow NS phase between the N and NF phases, i.e., the phase sequence on cooling is N-NS-NF. Apart from the initial studies that showed it (as well as some subsequent ones), this was proven by precision calorimetry in 2024: <https://doi.org/10.1103/PhysRevE.110.014703>.

Answer: We thank the referee for their comment and for pointing out this error. We have now corrected this section to better represent the phase sequence seen in RM734 along with additional references where required.

3. In Fig. 3 the authors should add a sketch of the polarizer/analyzer configuration somewhere.

Answer: We thank the referee for their comment and have added this to the figure along with the associated caption.

4. With regards to the dielectric measurements, in line 140 authors say that they use 9.7 micron-thick cells, while in line 204 they say 5 microns.

Answer: We thank the referee for spotting this and have corrected the text as new measurements have been made. The dielectric studies presented in current version of the manuscript were all done at the same measuring condition, using 20 micron-thick cells with gold electrodes and no surfactant layers were used and so this has been corrected.

“Investigating room temperature ferroelectric nematogens and their structure-property relationships” *review comments*

Comments on the main text page by page

Some individual items needing attention are highlighted.

Page 1

Page 2 Abstract

“... allowing for a detailed analysis of the structure-property relationships”

What is the journal policy about use of Arabic numerals vs. numbers “written out in full”?

The journal will decide this grammar issue.

Examples:

characterization of twelve

All seventy of the new compounds

Some fifty-two could be supercooled

... and other similar instances in the manuscript

“Only one pure compound to date has exhibited the N_F phase at this low temperature, UUQU-4N...”

This blanket statement above about UUQU-4N is found in the abstract (and thus also without immediate elaboration or citation). Are there no other single pure compounds in the literature in addition to UUQU-4N with such a low temperature N_F phase (or at least at or near ambient)? Perhaps some literature claims are incorrect (or actually do not apply to single component systems or fail for some other reason?). Single pure oligomers should be considered (even if they may otherwise ultimately fail somehow). Please elaborate on the claim made in the abstract about the uniqueness of the low temperature phase of UUQU-4N.

Some simple Google searches turned up some *possibilities* for published at or near room temperature N_F liquid crystals. What about:

<https://doi.org/10.1039/D5TC02738A>

Room-temperature ferroelectric nematic liquid crystals: the influence of substituents on properties (cite and discuss this recent publication which may postdate submission of this MS)

[DOI:10.1002/advs.202414317](https://doi.org/10.1002/advs.202414317)

Ferroelectric Nematic Liquid Crystals Showing High Birefringence (mixtures vs. single component?) currently citation 61

<https://arxiv.org/pdf/2309.14161>

Room-temperature ferroelectric nematic liquid crystal showing a large and divergent density (mixture?) not currently cited?

<https://doi.org/10.1080/02678292.2024.2345214>

A ferroelectric nematic liquid crystal vitrified at room temperature

“... phase behaviour leading to an N_F phase from 63°C down to room temperature.”

not currently cited?

<https://www.tandfonline.com/doi/full/10.1080/02678292.2021.1921867>

Ferroelectric nematic phase at and below room temperature

“We report a liquid crystal **1** that shows ferroelectricity at room temperature”

currently citation 18

<https://doi.org/10.1021/jacs.1c09594>

How Far Can We Push the Rigid Oligomers/Polymers toward Ferroelectric Nematic Liquid Crystals?

currently citation 36

<https://doi.org/10.1021/acs.macromol.1c00864>

Polar Liquid Crystalline Polymers Bearing Mesogenic Side Chains with Large Dipole Moment

currently citation 38

This is not a definitive search. The reviewer does not have access to some full articles. Some of these articles may not be relevant. In any case, it is worthwhile to validate the claim.

Thus far it has certainly been a challenge to identify single component materials which have an N_F phase at or near ambient temperature. The search for such materials is worthy but ultimately is this not the pursuit of the idiomatic sacred cow in a liquid crystal context? Mixtures are commonly (if not invariably utilized) in lieu of unitary composition to attain the requisite wide variety of function required for liquid crystal applications. In this regard, is the examination of some mixtures of these new compounds a worthy investment of time and effort? It appears mixtures with RM734 have been discussed here but only in the context of assisting phase identification (Figure 4 of the manuscript). Please elaborate.

Page 3

A dioxane ring (at least in the context of DIO) is not a *lateral* group since it is not attached to the *side* of the molecule. In DIO the 5-alkyl-1,3-dioxane ring attached at the 2-position is a *terminal* group.

Page 4

Seventy new ferroelectric nematogens in the manuscript ... a substantial number added to the existing about two hundred and fifty. Common feature is a lateral alkoxy chain.

The discussion provided about the structure variations is well managed given the scope of structures found in the literature.

This manuscript follows from previous papers and previous strategy from this group (including references 19, 20 21,25, 31, 32 ... others?).

Page 5

Consider adding “previously reported” to the caption of Figure 1.

Figure 2 shows the “reported compounds” in this new manuscript which number $2 \times 8 + 10 \times 7 = 86$ compounds (of which seventy are ferroelectric)?

Would it be helpful at this stage to explain up front the differences between Figure 1 and Figure 2? For example, the introduction of the 4-methyl group instead of a methoxy group is quite comprehensive in *A.m* through *H.m* while *E.m* through *L.m* have two lateral alkoxy groups not seen previously in Figure 1.

Page 6 – Page 8

The manuscript is devoted and limited to the modification of the system

C₆H₅-COO-C₆H₄-COO-C₆H₄-NO₂

involving replacement of various hydrogen atoms with a selected limited number of substituents Me, OMe, OR (longer linear alkoxy), and F

in different combination, number and locality.

These few variables already deliver a huge matrix of options which are being systematically addressed (and obviously numerous other such modifications are conceivable).

The synthesis schemes pages 6 through page 8 are clear and the experimental methods pages 8 through page 9 are clear. The synthesis steps are not described in much detail (and has already been done in earlier related publications). The variability in esterification yields are worthy of some comment.

There appears to be a change in nomenclature/style between Figure 1 and Figure 2 and the style found in Scheme 1 through Scheme 3. In Figure 1 and Figure 2 “OC_mH_{2m+1}” appears for the lateral alkoxy groups while in Scheme 1 through Scheme 3 “Om” is used. The latter style seems unnecessary and is somewhat unorthodox. Please comment on style consistency (and even as it pertains to style in earlier publications cited).

Page 8- Page 9 Various Measurements / Modeling

This subsection is fine except the absence of any polarisation measurements is surprising.

This group has reported polarization measurements in some previous publications (ref 20, ref 21). The reviewer has no reason to doubt the existing phase assignments made in the absence of polarization measurements. Nonetheless, polarization measurements constitute a main feature in most discussions about all ferroelectric substances. Please comment on the absence of at least some representative polarization measurements.

Page 9 Results

Eighty-six ferroelectric nematogens reported and twelve of them are previously reported.

Only three nematogens are reported to possess a standard nematic phase in addition to the ferronematic phase. Is there any benefit to having this adjacent nematic phase (such as for alignment purposes)? Please elaborate about the impact of an adjacent nematic phase on the utility of the nematic ferroelectric phase.

Page 10 – Page 12

The high viscosity of (some or all?) materials at low temperatures is noted in the context of issues with texture assignment. This appears to be the only mention of viscosity issues of these materials in the entire manuscript. How pervasive is this viscosity issue? Do the “low temperatures” mentioned here include room temperature? Are there any performance consequences associated with the high viscosity (e.g., switching dynamics)? Some applications of NF materials may not be impacted by high viscosity (or could even benefit from it?). So, while the desired N_F phase exists at lower temperatures in many of these new materials, are there some

performance tradeoffs due to high viscosity near ambient temperatures in these new compounds which should be considered? Phase assignments reported on the high viscosity materials required the use of a binary mixture. Are mixtures also going to be necessary in some potential application(s)? These issues are overlooked so far and require elaboration (and, if the issues really are troublesome, then solutions are required).

Oligomers and polymers of p-hydroxybenzoic acids have been widely studied. The viscosity of such polyesters is a critical processing parameter. Perhaps this is a place to learn something about the influence of substituents on phase behavior and viscosity. It is not clear whether or not any existing information is relevant to ferronematic materials, but it seems worthwhile to look at since significant effort has gone into the understanding and optimization of LC polyesters. The reviewer did some quick searching about relationships between functionalization of these polymers and viscosity but did not turn anything very relevant up. Sorry. Such information must be available.

Page 13

Why is reference 18 cited in relation to some forthcoming discussion?

Based on the discussion on this page about the glass transition the reviewer anticipated seeing an example of a DSC with this feature. While information about this may be included somewhere in Tables S28-S29 for the benefit of readers unfamiliar with this issue can you also show an actual DSC trace with this feature?

Page 14 - Page 15

Dielectric permittivity data is provided to support assignment of the N_F phase (and, as noted, in spite of associated controversy about anomalous large values). As already noted, there is a conspicuous absence of any polarization measurement in this manuscript that often is a critical component of numerous other papers on this topic. What is lost (if anything) by not having some polarization data to examine at this stage?

Page 16

Are citations available to support the statement “conventional theory of the nematic phase for which a decrease in shape anisotropy leads to a general destabilisation of the phase”? If this statement is accurate and can be applied broadly (including ferronematics) some additional discussion would be helpful. Please provide background on this.

In Figure 7 “Dependence of the transition temperatures on the number of carbon atoms in the lateral alkoxy chain” is presented. The materials have multiple phase changes. Specifically, which phase change(s) are plotted? Can this be succinctly indicated in the figure or in the caption?

Page 17 – Page 18

Some of the structure variations of substituents on RM734-like substances found in research to date is discussed. Absent from the discussion are reported substitutions of heterocycles for the benzene rings. Shuffling some (hetero)atoms *in* the rings is an equally legitimate method to optimize ferroelectric behavior (existence, magnitude, temperature range, etc.) as the same activity *on* the rings. For example, see:

<https://doi.org/10.1039/D4TC02851A>

Sulfur-based ferroelectric nematic liquid crystals (thiophene)

<https://doi.org/10.1080/02678292.2022.2145380>

Supramolecular ferroelectric nematic materials (pyridine)

doi.org/10.1002/cplu.202300726

The Emergence of a Polar Nematic Phase: A Chemist's Insight into the Ferroelectric Nematic Phase
Currently citation 14 (compound 83, pyridine)

Page 19

In Figure 8, where is the arrow indicating the positive direction of the calculated dipole moment mentioned in the caption?

Page 20

Discussion is presented in this section about the comparison of methoxy and methyl terminal groups. The key difference between the methoxy group and a methyl group is an oxygen atom (two "large" atoms vs. one). Simple structure changes have less than simple outcomes.

Page 21

Using 5-6¹⁹ from **the** literature

The 5-6¹⁹ is not clear to the naïve reader (but may be to the authors familiar with this nomenclature). Should this read 5-6¹⁹ ... italicized consistent with Figure 1?

"... there are still limitations in our understanding of the interplay between shape and electronic effects in driving N_F phase formation" **AGREED**

"Our next target is therefore reduction of the melting points in these materials through further structural modifications to attain highly desirable enantiotropic, room-temperature ferroelectric nematogens suitable which would be suitable for use in devices." **GOOD LUCK** Enantiotropic N_F behaviour is rare or absent amongst the new compounds in the manuscript. Does the basic three ring diester system examined here provide adequate "bandwidth" for delivering the desired enantiotropic N_F phase (including at or near ambient temperature)?

Page 22 – Conclusions

"This shows that by carefully using specific structural changes it may be possible to tailor the transition temperatures of these materials. Indeed, if the melting points these materials could be lowered it does raise the exciting possibility that a room temperature enantiotropic N_F material is very much feasible." **AGREED as a concept**

It appears likely that further improvements in phase behaviour will be found in the RM-734 class of compounds. But here is a strategy consideration: perhaps there is some currently unidentified system which has more "dynamic range" and is also easier to manipulate to deliver lower temperature ferroelectric phases along with other acceptable physical properties. So, while there is value in digging in deeply within a specific format (e.g., RM734 type substances) there is also the risk that it may not end up being the most expedient or effective format to modify.

What say ye?

Comments on supplementary information

Generally, the supplementary information section is meticulously and thoroughly prepared and presented. This is much appreciated.

The entire SI technical data section was NOT examined with a fine-tooth comb.

SI Page 8

Colourless oil. Yield: 4.32 g, 91 %. RF: 0.33 (100% dichloromethane)

Table SI3, SI5 and others

In this study DCC is used most often for Steglich esterifications. The yields for various DCC esterification reactions vary quite a bit (here and elsewhere) and are sometimes quite low. Is there some reason for this? Is there any rhyme or reason (intrinsic to the reaction itself or variability in the workup/isolation)? The yields are what they are ... just wondering about this in case there may be some remedy to suggest.

There is not much in the way of synthesis methodology discussed in the main text but hopefully this issue can be discussed in the main text or otherwise in the supplementary information.

Artwork / Nomenclature for Synthesis Schemes 1-4

Some of the artwork/nomenclature utilized is unconventional. For example, in Scheme 1 compound (1) has a substituent at the 2-position which is “Om” where “m” is the number of carbon atoms in the chain. Typically, in usual descriptive parlance, this transformation would be indicated with the new 2-substituent being $\text{OC}_m\text{H}_{2m+1}$. So, this is not wrong ... just unconventional and different from style used elsewhere.

This issue was mentioned earlier about the Synthesis Schemes in the main text.

Table SI9.1, Table SI9.3

The esterification reagent is now EDC. Is there some rationale to use one carbodiimide instead of the other (here and elsewhere)? Again, while there is not much synthesis discussion in the manuscript any insights like this could be valuable.

Additional Data

It would be very helpful to add general structures at the top of Tables SI28 – SI39 so the reader can be spared seeking this info elsewhere.

There are eighty-six compounds found in Tables SI28 – SI39 (of which seventy are new?). Is it possible to provide some indication (perhaps a citation) for the compounds which have been previously published? This additional bookkeeping would help clarify progress in this class of compounds. On page nine of the manuscript the following twelve entries have been identified as already reported: A.1,²⁵ B.1,²⁵ C.1,²⁵ D.1,^{24,25} E.1,²⁵ F.1,²⁵ G.1,²⁵ H.1,²⁵ I.1,²¹ I.2,²¹ J.1,²¹ J.2.²¹

The inclusion of this information in the Tables would be helpful.

In a number of places in the manuscript there are mentions of “x of this flavor” and “y of this flavor”, etc. The categories and populations are of interest. Can they somehow be summarized in the existing tabulation?

Summary

This manuscript is from a joint project at the University of Aberdeen and the University of Warsaw. This manuscript follows from directions and successful outcomes which are described in a number of significant previous publications.

Generally speaking, a strength of the manuscript and its supporting information is that they are meticulously prepared.

A strength of the manuscript is the sheer number of individual new substances described and the discovery that a significant fraction is ferronematic.

A current weakness of the manuscript is the lack of additional characterization data (e.g., polarization, x-ray). Some earlier publications by this joint project include polarization measurements. Comprehensive physical measurements on such a large number of new compounds are not practical and are not required, but at least some representative measurements would be instructive and supportive. Of course, an argument might be made that they may not. Please reply.

The current manuscript focusses heavily on the behavior of individual compounds. Knowledge about the physical properties of single compounds is valuable but a practical solution ultimately lies in the formulation of mixtures. Should this mixture approach be applied in parallel to the development of individual compounds? What is the best set of physical properties to measure? These are some generic questions.

A current and primary weakness of the manuscript is the paucity of discussion about the origin and/or impact of the high viscosity observed in some of these new N_F materials. The reviewer is concerned this feature may impede ultimate practical applications of this class of materials. It appears that structure modifications implemented to lower the NF phase temperature range have been successful, but they may well also bring about an increase (or at least permit revelation) of a high viscosity régime near room temperature. Better understanding of the structure/viscosity relationship will turn iterative progress into a breakthrough which will pay off here and could likely pay off for other materials classes as well.

Earlier, the understanding and development of simple nematic liquid crystals went through a protracted period dealing with viscosity issues. It is no surprise the same issue is relevant with ferronematics. The methods and mechanisms leading to improvement in the simple nematics may or may not apply to the ferronematics. Please address the viscosity issue.

The reviewer looks forward to hearing back from the authors. Does some existing information assist in solving about the viscosity issue? What appropriate additional characterization will assist to obtain a solution to the viscosity issue?